# Multiphase detection of crucial biological amines using a 2,4,6-tristyrylpyrylium dye
Shivani Tripathi, Banchhanidhi Prusti ⓘ & Manab Chakravarty ⓘ ✉

The strong electrophilicity of arylpyrylium salts was recognized for the colorimetric detection of vital amine analytes, limited to ammonia or methylamines and putrescine as biogenic amine. This report presents conformationally twisted, electrophilic triphenylamine-linked *2,4,6-tristyryl*pyrylium *salt* **PyTPA** as a single dye to sense various aliphatic/aromatic biogenic amines, nicotine, and guanidine rapidly in nanomolar concentrations. This unexplored styrylpyrylium design offers specific electronic conjugations, steric/geometric constraints with hydrophobicity, and decent thermal/photostability, facilitating precise diverse amines detection in unique fashions. The deep-violet solution/solid dye responded remarkably at 298 K with quick decoloration against putrescine, cadaverine, spermidine, spermine, histamines, serotonin, and 2-phenylethylamine. Further, this dye could detect nicotine at 313 K and guanidine at 298 K distinctively with diminished absorption and unexpected red-shifted emission enhancement. Variation in mechanistic path is recognized in detecting amines holding mono/di-NH$_2$ groups and short/ long alkane chains, elucidated by mass, $^1$H-NMR, FT-IR, SEM, PXRD, and XPS studies. The notable detection of these biogenic amines in different phases is employed for onsite applications to detect fresh chicken and tuna. Nicotine in natural tobacco leaves was identified. Such pyrylium salt provides promising advancements in this class of molecules in detecting diverse biologically significant amines.

Since the discovery of pyrylium salt in the realm of organic chemistry, it has been extensively utilized in various applied fields such as photocatalysis[1], light emitter[2], chemosensing and functionalized metallo-supramolecules[3–6]. The permanent positive charge on the oxygen atom in this pyrylium core makes this core highly susceptible to a wide range of nucleophiles. In particular, this feature offers colorimetric and fluorometric detection of analytes such as amines[7], carbonates[8] that are related to environmental monitoring and industrial developments. Among different nucleophiles, various amines are majorly employed to create pyridine heterocycles or pyridinium salts through favorable heteroatom replacement ability[6,9]. The pyrylium-based polymers/frameworks are well-established for efficient colorimetric sensing[8]. Conversion of pyrylium cation to pyridine or pyridinium salt by using amine will influence the photophysical features of the dye and thus can offer colorimetric or fluorometric detection. Earlier reported pyrylium salts **NPY** and **EPY** (Fig. 1) were realized for fluorometric detection of essential amines, specifically methylamine[10], ammonia,[11,] and partly biogenic amines (only putrescine)[12].

The biogenic amines are responsible for essential metabolic pathways, yet cause toxicity after reaching a higher concentration (>20 mg/kg)[13]. Therefore, the amount of biogenic amines originating from microbial decomposition of amino acids measures the freshness of food[14]. Hence, easy

detection of biogenic amines is highly desired to recognize fresh food, specifically perishable protein-rich food items such as fish, meat, and paneer. Among various expensive analytical methods for biogenic amines detection[14–17], colorimetric/fluorimetric analysis is the most demanding due to its higher accessibility, sensitivity, simplicity, and cost-effectiveness. Notably, pyrylium dye **NPY** was not explored for multiphase detection of important aliphatic or aromatic biogenic amines. Again, a mechanistic understanding was restricted to only ammonia, not with biogenic amines. Another pyrylium fluorophore **EPY** (Fig. 1) was introduced to detect 1°/2°/3° amines with a primary focus on *methylamine*. However, no biogenic amines detection was attempted using this probe.

Considering this, we believe that pyrylium ion will have the potential to detect various biologically relevant amines, including biogenic amines (aliphatic, aromatic, neurotransmitters), nicotine, and guanidine, through visually detectable approaches. However, the steric selectivity and reactivity of pyrylium salt towards nucleophiles are concerns and dependent on the type and nature of the substituents[18]. In this direction, with our recent interest in biogenic amines detection[19–21], we herein report a class of 2,4,6-*tris*((*E*)-4-(diphenylamino)*styryl*)pyrylium (**PyTPA;** Fig. 2), a different thermally and photochemically stable design compared to related arylated pyrylium cores.

Department of Chemistry, BITS-Pilani, Hyderabad Campus, Jawahar Nagar, Shamirpet, Hyderabad, India. ✉e-mail: manab@hyderabad.bits-pilani.ac.in

**Fig. 1 | Arylated Pyryliums.** Earlier reported pyrylium motifs for amine sensing and their drawbacks.

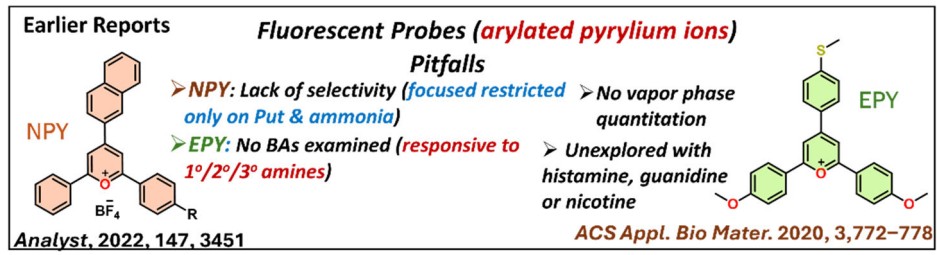

**Fig. 2 | Summary of this work.** The current report on the probe **PyTPA** with key benefits.

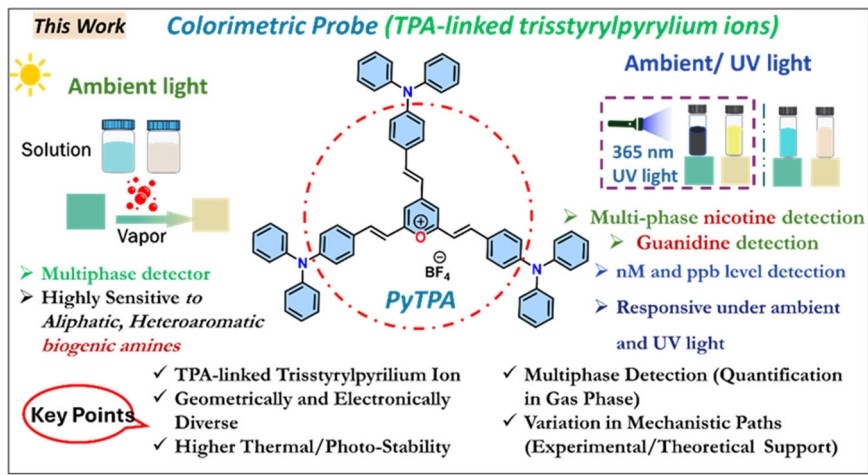

Based on the intriguing variation in crystal structure and molecular packing of simple *tri*-styrylpyrylium salt with *tri*-styrylpyridine[9], the change in the photophysical feature of **PyTPA** was deliberated upon exposure to various amines. It would respond rapidly to key amines, primarily biogenic amines, after converting into corresponding pyridinyl salt and exhibiting absorption or emission switching. Unlike the previous reports, **PyTPA** is realized to be a non-fluorescent bluish dye that responds with significant absorption reduction against aliphatic and aromatic biogenic amines, including histamine and other neurotransmitters in an nM concentration range. Even though multiple colorimetric/fluorometric probes exist for aliphatic biogenic amines, but suitable probes such as serotonin and histamine are sporadic[22–24]. Moreover, histamine is majorly responsible for scombroid food poisoning[25]; therefore, handy histamine detection has extensive demand worldwide[26]. Further, **PYTPA** could also detect other vital natural amines such as guanidine and nicotine by diminishing the absorption and detectable enhancement of emission intensity. Of note, guanidine responded against the probe at 298 K, while nicotine was receptive only after heating at 313 K. This colorimetric detection was prominent in the solid and solution states of the probe with solution and vapor states of analytes, wherever applicable with higher vapor pressure at 298 K. Interestingly, the reaction between **PyTPA** and mono/di amines [NH$_2$-(CH$_2$)$_n$-NH$_2$] with short alkane chain (n ≤ 4) produce only monopyridinium salts (MPS), whereas diamines with long (n > 4) alkane chains produce a mixture of MPS and bis-pyridinium salts (BPS). Notably, guanidine also formed the MPS, but nicotine is sensed by the strong intermolecular interactions with dye at 313 K. The mechanistic variations are supported by mass, FT-IR, SEM, and XPS studies. The selectivity with amines is elucidated by deciphering the role of hydrophobic TPA-linked styryl units. Finally, this probe was utilized to monitor the fish spoilage and detect nicotine from tobacco leaves. Many sensors were previously recognized individually to detect biogenic amines[27,28], nicotine[29,30] and guanidine[31,32], but a single probe capable of detecting and discerning these vital amines is unique and well deserved. Such colorimetric detection would prevent the generation of electronic waste as an alternative tool.

## Result and discussion
### Design and synthesis of the probe

The hitherto reported 2,4,6-tristyrylpyrylium tetrafluoroborate structure owns a planar molecular structure with a subtle distortion where iso-electronic analog 2,4,6-tristyrylpyridine holds a distorted geometry[9]. Further, the molecular alignment in the crystal packing also differs between this styrylpyrylium and styrylpyridine[9], resulting in a difference in the dipole moment that might direct the photophysical features. Herein, triphenylamine (TPA)-linked pyrylium cation is designed for the following benefits: (a) easy accessibility and synthetic versatility[33] (b) good thermal and photostability[34] (c) high optical absorption coefficient[35] (d) required solubility (10 µM) in various organic solvents[36] (e) film-forming ability[37]. and (f) high sensitivity with its high hole mobility[38]. (g) the propeller-shaped TPA core, proficient of adopting geometrical and packing diversity[39] (h) enhancement of hydrophobicity with multiple phenyl rings[40]. The aldol reaction between 2,4,6-trimethylpyrylium tetrafluoroborate (PY) with TPA-aldehyde could afford shiny deep violet **PyTPA** in 85% yield (Scheme 1). This **PyTPA** presumably offers a different steric environment and specific cavity for detecting selective amines. The styryl link with a TPA core will further control the reactivity of nucleophiles and tune selectivity towards specific amines compared to aryl-linked counterparts. Moreover, such TPA substitution would offer more hydrophobicity and different steric constraints, resulting in selectivity towards specific amines. The TGA (thermogravimetric analysis) of **PyTPA** shows thermal stability up to ~250 °C (Fig. S1a in Supplementary). The thermal stability of earlier reported **EPY** and **NPY** were not tested. The photostability of **PyTPA** (10 µM in MeCN) was tested by continuous exposure to room light (intensity ~300 Lux) for one day, where absorption remained unchanged for more than a day, but a marginal reduction in the absorption was perceived on the 2nd day (See absorption spectra Fig. S2a, b). We also showed that the

**Scheme 1 | Synthesis of the dye**. Synthetic scheme to access **PyTPA** with concept design.

- ➢ Easy accessibility and synthetic versatility
- ➢ Good thermal and photostability
- ➢ High optical absorption coefficient
- ➢ Film-forming ability
- ➢ Notable structural design with high hydrophobicity

**Fig. 3 | Photophysical properties of the dye. a** UV-vis and **b** emission spectra ($\lambda_{ex}$ = 410 nm) of **PyTPA** in 10 μM; inset image is **PyTPA** under room light and 365 nm UV lamp in different solvents; (i) n-Hexane (ii) Toluene, (iii) 1,4-Dioxane (iv) Tetrahydrofuran, (v) Chloroform, (vi) Dichloromethane, (vii). Acetonitrile, (viii) Dimethylsulphoxide (ix) Methanol (x) Water. **c** TD-DFT calculated depiction of HOMO-LUMO transition in **PyTPA** molecule.

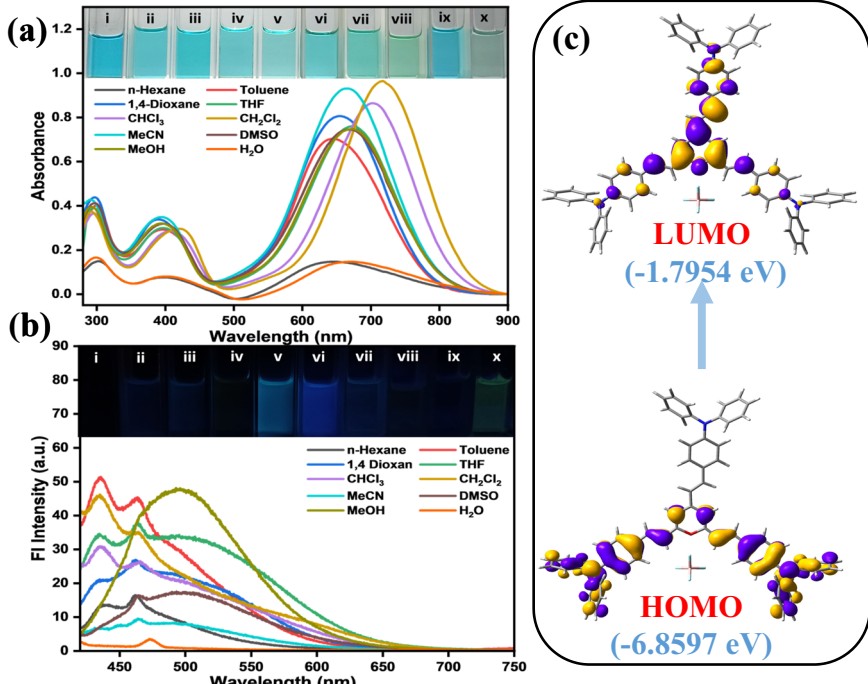

**Photophysical properties of the dye and the response against amine solution**

The synthesized **PyTPA** was soluble in various organic solvents of different polarity. The absorption spectra of this compound (10 μM) showed broadly structured signals at $\lambda_{max}$ = 400 nm and 663 nm, which could be attributed to the π-π* and charge transfer (CT) band, respectively (Fig. 3a). The CT band possibly arises from the electron-rich triphenylamine core to the positively charged pyrylium ring. To confirm the electronic structure, the molecular structure was optimized through density functional theory [DFT/ωb97xd/6-31+g(d,p)] (Fig. 3c). The electronic clouds are localized within the triphenylamine cores in the highest occupied molecular orbital (HOMO), whereas the lowest unoccupied molecular orbital (LUMO) is localized on the pyrylium core. The optimized structure was further analysed using time-dependent DFT (TD-DFT) calculations, employing the same level of theory and incorporating the CPCM model in acetonitrile. The

TD-DFT data shows a strong absorption band at $\lambda_{max}$ = 655 nm, majorly due to the HOMO → LUMO transition (Fig. S3, Table S1). The primary charge transfer band at 663 nm (molar absorptivity, ε ~ $10^4$ $M^{-1}$ $cm^{-1}$) originates this intense blue coloration (Fig. 3a). However, this compound with CT feature is feebly emissive (almost non-emissive) in most of the solvents (Fig. 3b), possibly the multiple phenyl rotors promote the relaxation through nonradiative paths. Some detectable emission is recorded in chlorinated solvents $CH_2Cl_2$ and $CHCl_3$ possibly by creating some specific noncovalent interactions. Therefore, this probe is mainly employed as a colorimetric probe, not a fluorescent one.

Upon treating **PyTPA** in MeCN (10 μM, ε = 8 × $10^4$ $M^{-1}$ $cm^{-1}$) with various amine solutions [10 μM in dimethyl acetamide (DMAc)], absorption and emission spectra were recorded (See Figs. 4a and S4a) at 298 K. Initially, absorbance remained almost unchanged upon addition of caffeine, di-iso(propyl)ethylamine (DIPEA), $NEt_3$, ammonia, and nicotine. With our primary focus on various biogenic amines, 1,3-diaminopropane (1,3-DAP; analogous to biogenic amines), 1,4-diaminobutane (1,4-DAB or putrescine, PUT), 1,5-diaminopentane (1,5-DAP or cadaverine, CAD), 1,6-diaminohexane (1,6-DAH), spermine, spermidine, guanidine, and histamine, 2-phenylethylamine (2-PEA) were quickly responsive with a drastic

absorption only reduces upon adding CAD (Fig. S2a). Additionally, photostability was studied by exposing the dye to sunlight for 3 days (intensity 400–10,000 lux), where negligible decrement of absorbance was observed (Fig. S2c, d).

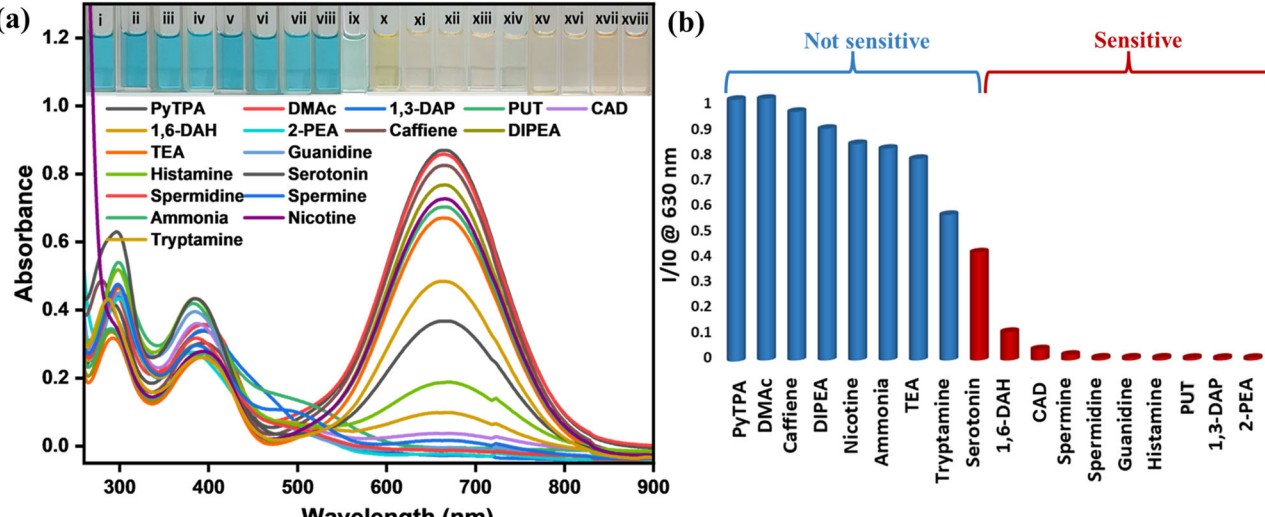

**Fig. 4 | Colorimetric responses against various amines. a** UV-vis spectra of **PyTPA** (10 µM in MeCN) upon addition of respective amines (10 µM in DMAc) at 298 K [guanidine, serotonin, and histamine were dissolved in water]. Inset presents visually dateable images of **PyTPA** with different amines under ambient light; (i) **PyTPA** (10 µM), (ii) DMAc, (iii) caffeine, (iv) DIPEA, (v) nicotine, (vi) ammonia,

(vii) triethylamine, (viii) tryptamine, (ix) serotonin, (x) 1,6-DAH, (xi) CAD, (xii) spermine, (xiii) spermidine, (xiv) guanidine, (xv) histamine, (xvi) PUT, (xvii) 1,3-DAP, (xviii) 2-PEA. **b** Bar diagram representing the sensitivity of **PyTPA** for various classes of amines, where the absorbance of **PyTPA** (10 µM in MeCN) at 630 nm is reduced upon adding respective amines.

absorbance reduction (See Fig. 4a). A partial (40–50%) absorption reduction was noticed with tryptamine and serotonin (Fig. 4b). The fluorometric detection route for these amines was selectively adopted here due to weak emission and subtle responses (Fig. S4b). Surprisingly, guanidine showed a noticeable emission intensity enhancement at 298 K, indicating the reaction between the dye **PyTPA** and the strong base guanidine that led to a unique product holding better emission efficiency than **PyTPA**. Thus, guanidine could be detected by both colorimetric and fluorometric routes.

We further tested other various amines that include primary amines [ethylamine, n-butylamine, benzylamine, and aniline], secondary amines [diethylamine (DEA)], tertiary amines [DIPEA], and amines holding both tertiary and primary amines, i.e., *N, N*-dimethyl-1,3-propanediamine (DMPA). We found no change in the absorption for almost all the amines; only DMPA showed a slight diminution in the low-energy absorption because it is also a diamine where the interaction happened from one amine end, and the attack can happen from the other amine part. However, this reduction is not very significant in comparison to biogenic amine. Further, the comparison was made with alcoholic and phenolic nucleophiles by using methanol, 1,4-butanol, 4-aminobutanol, ethylene glycol (EG), phenol, catechol, resorcinol, and even dopamine, exhibiting only a slight change in the absorption (Fig. S5).

Furthermore, we also have investigated the response of the dye upon mixing different amines. In the beginning, upon the addition of 10 nM PUT (in DMAc) to 10 µM of PyTPA (in MeCN), the absorption at 665 nm reduces (~23% of the initial absorption) even with such a small amount of PUT. Next, 10 nM CAD (DMAc) also displayed a similar reduction in absorbance. To compare this reduction, 10 nM DIPEA (in DMAc) was added, and only 15% of the absorption lessening was revealed, as observed in the initial screening. Later, 20 nM of spermidine (DMAc) was treated, and a 77% reduction in the absorbance was observed (Fig. S6). Thus, this dye exhibited the potential to detect important BAs even in the mixture of various amines.

### Responses against aliphatic biogenic amines and their analogues

Next, the absorption spectra of **PyTPA** were separately recorded upon slow and gradual addition for all the aliphatic diamines (1,3-DAP, PUT, CAD, 1,6-DAH, spermidine, and spermine). We monitored the absorption at 630 nm ($\lambda_{max}$), which diminished rapidly upon treatment with all these

aliphatic biogenic amines (Fig. 5a–f). The isosbestic point around 500–530 nm in the absorption was noticed for all the cases, indicating the formation of a new species.

As the change in weak emission was meagre, we refrained from further discussions on emission spectra (Fig. S7a–f). The limit of detection (LOD) was calculated using 3.3σ/k (k = slope and σ: standard deviation) and appeared to sense these biologically relevant amines as low as nanomolar (nM) level (37–100 nM) (Fig. 5, Fig. S8a–f, Table S2a–f).

### Responses against other neurotransmitters-based (hetero)aromatic biogenic amines

The 2-PEA, histamine, and serotonin are neurotransmitters and are noticed in various foods and beverages. Maintaining their concentrations in food is desired as they bring impactful toxicity to humans[41,42]. Only limited probes are recognized to detect these particular biogenic amines[23,43]. Therefore, effective detection of these biogenic amines illustrates an emerging demand. The absorption change was very similar to the aliphatic biogenic amines, displaying a complete drop upon gradually adding these neurotransmitter-based biogenic amines (Fig. 6a–c). The calculated LOD was in the nM range, as specified in the insets of Fig. 6a–c (Table S2g–i). The observed isosbestic points specified the presence of new species. The effect of these amines on emission spectra was also detected with a slight intensity enhancement (Fig. S8g–i). However, the prominent impact was noticed only in the case of guanidine and nicotine.

### Responses against guanidine and nicotine

Apart from biogenic amines, we also extended the applications with two other crucial bases, i.e., guanidine and nicotine. Moreover, the initial studies with guanidine were promising. It is pertinent to mention that the strong base guanidine is a vital building block in organic chemistry and has various industrial applications[44]. The chemical reactions of guanidine with pyrylium salt were extensively studied, but photophysical features were never explored. The colorimetric detection of guanidine is sporadic[45–48]. There could be many other vital amines under this category, but we restrict only guanidine to find further scopes. Additionally, nicotine, a naturally occurring alkaloid, is present primarily in tobacco. Moreover, nicotine has a stimulating effect on the central nervous system, leading to various harmful effects concerning pulmonary and cardiovascular disease[49–51]. Thus, handy detection for both guanidine and nicotine brings significant impact. The

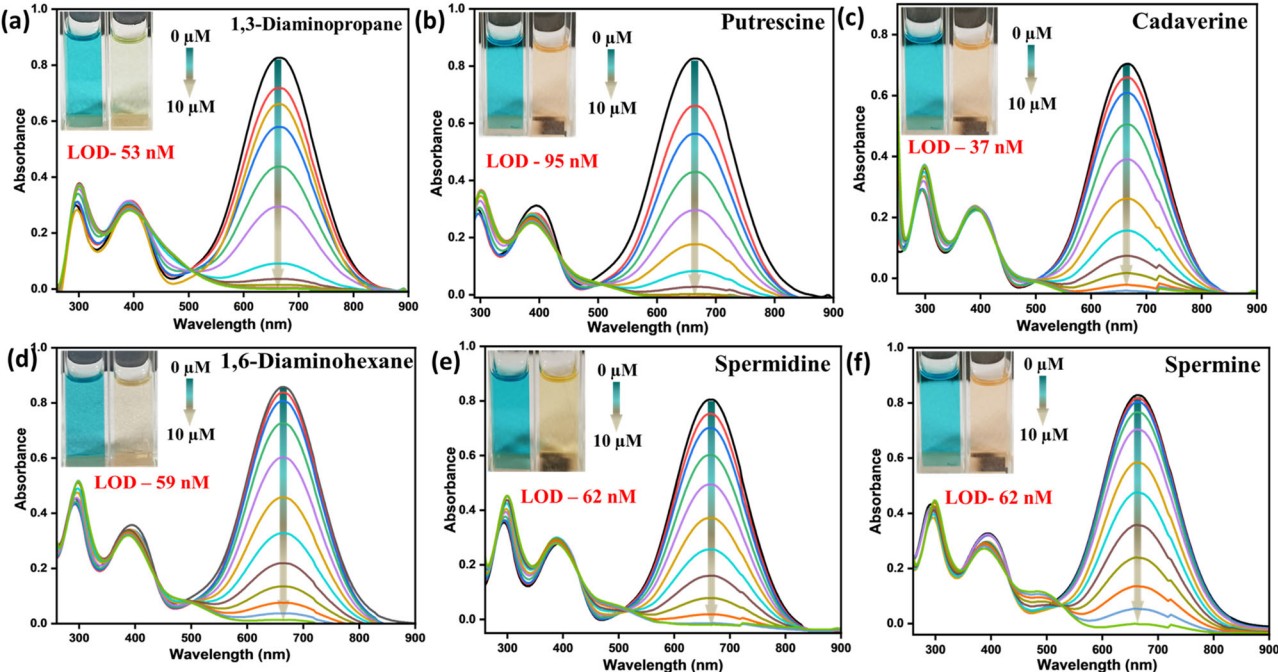

**Fig. 5 | Change in the UV-vis spectra of PyTPA (10 μM in MeCN) at 630 nm upon gradually adding respective amines. a** 1,3-DAP, **b** PUT, **c** CAD, **d** 1,6-DAH, **e** spermidine, **f** spermine, Inset: photograph of the **PyTPA** (10 μM in MeCN) before and after amine addition in ambient light.

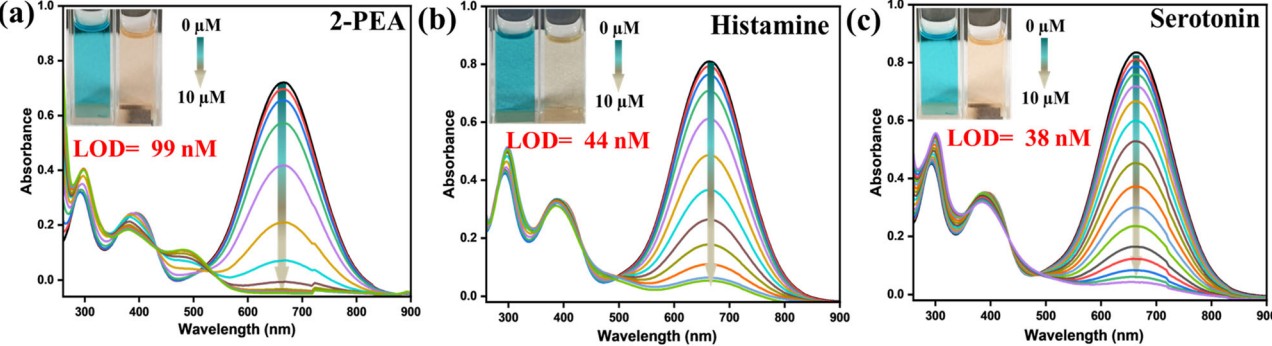

**Fig. 6 | Absorbance changes upon the addition of neurotransmitters.** Change in the UV-vis spectra of **PyTPA** (10 μM in MeCN) at 630 nm upon gradual addition of **a** 2-PEA, **b** histamine, and **c** serotonin, [Inset]: photograph of the **PyTPA** (10 μM in MeCN) before and after amine addition in ambient light. The histamine and serotonin salts were used in water after neutralization.

absorption was promptly reduced with the incremental addition of guanidine solution (Fig. 7a), and the measured LOD lies in the range of 119 nM. As stated before, fluorescence intensity was enhanced almost 25–30 times upon incremental addition of guanidine till $10 \times 10^{-6}$ M, with 25 nm bathochromic shift cyan-green emission [relative quantum yield ($\phi_f$):18.20%], indicating the emissive nature of the newly formed species. Unlike other amines, nicotine was responsive against this dye only at 313 K (Fig. 7b), possibly due to the absence of -NH functionality. The absorption was drastically reduced with LOD with a slightly higher range of 196 nM LOD (Fig. S8j, k, Table S2j, k). Such a difference in LOD from other amines indicates weaker reactivity of nicotine than others (*see later*). In addition, ~20 times emission enhancement with a 34 nm bathochromic shift (green emission) (Fig. S9a, b) was prominent [$\phi_f$:11.09%] in the emission spectra. The probe **PyTPA** interacted with nicotine and formed the complex in the ground state, which shows no absorbance at 630 nm but emits upon excitation at 410 nm. The bathochromic shift infers an improvement of π-conjugation of the dye after forming the **PyTPA**-nicotine complex (*vide infra*), possibly created through supramolecular interactions. The emission intensity enhancement for both guanidine and nicotine (Fig. 7c, d) specifies

attaining molecular rigidity to enable a radiative decay, as observed through excited state lifetime studies (*see later*).

Moreover, the absorbance responses at various temperatures were carried out for this dye against CAD as a representative example (Fig. S9c). Instantaneous changes with decolouration were noticed at different temperatures of −18 °C (255 K), 0 °C (273 K), and 20 °C (293 K), indicating the reactivity of the dye even at low temperatures.

**Paper strip fabrication and characterization**

The PyTPA ($10^{-3}$ M in $CH_2Cl_2$) was drop-casted on a Whatman filter paper (WP) (100% cellulose fiber) and dried at 298 K for 15 min to afford a cyan-green platform, The dried paper strip was further characterized by SEM, IR, and UV-Vis (Fig. S10a–c) spectroscopic methods before (**PyTPA@WP**) and after exposure to CAD vapors (**PyTPA@WP + CAD**). It was noted that **PyTPA** was absorbed into the pores and agglomerated on empty pores of paper, as noticed in the SEM images [Fig S10a(i-iii)]. Further, the IR spectrum of the **PyTPA**-casted paper strip (**PyTPA@WP**) demonstrated the appearance of 1544 cm$^{-1}$ (C = O$^+$), which was absent in WP. Upon exposure to CAD, the peak at 1544 cm$^{-1}$ was shifted to 1589 cm$^{-1}$ (C = N$^+$).

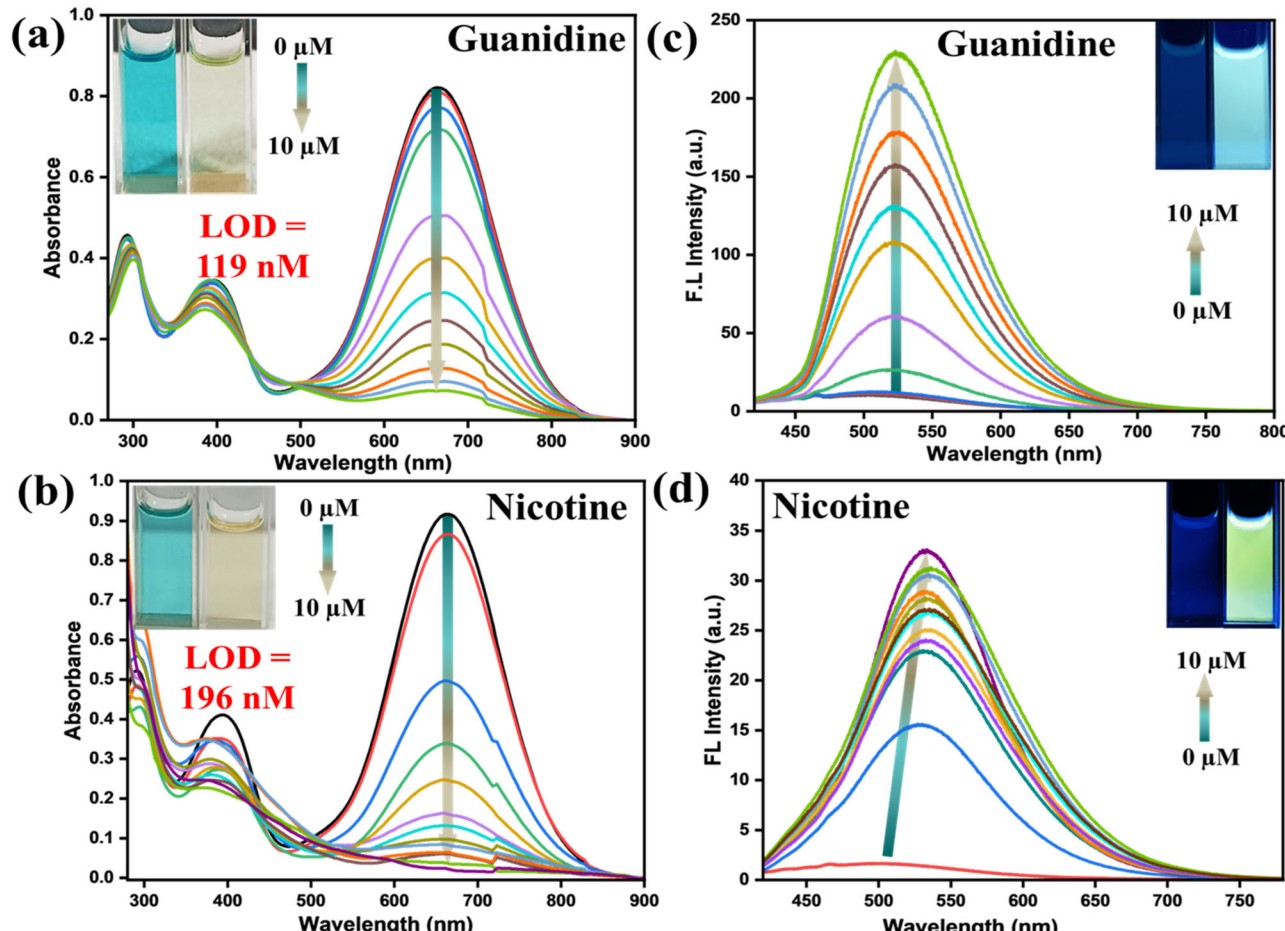

**Fig. 7 | Photophysical properties variations for guanidine and nicotine.** Change in the UV-vis spectra of **PyTPA** (10 µM in MeCN) at 630 nm upon gradual addition of **a** guanidine, and **b** nicotine (The guanidine salt was neutralized in water and used for this study), Change in the Fl intensity of **PyTPA** (10 µM in MeCN) by gradual addition of **c** nicotine (was heated at 313 K for 30 min, $\lambda_{ex}$: 410 nm), and **d** guanidine ($\lambda_{ex}$: 410 nm). Inset: photograph of the **PyTPA** (10 µM in MeCN) before and after the addition of nicotine and guanidine in ambient light and under UV light at 365 nm.

The measured UV-Vis spectra of **PyTPA@WP** displayed the expected absorption at 310, 430, and 660 nm, where the 660 nm signal completely disappeared on exposure to CAD vapor.

**Detection of biogenic amines in the solid state of the probe**
Solution-based detection is mostly suitable inside a laboratory, and the preparation and spillage of the solution need special attention. Detecting analytes using a thin film strategy is highly desired to make the process handier and operationally simple for on-site applications. Hence, the photophysical properties of the dye are deliberated to test upon the addition of the above-mentioned amines in solution and vapor state. The deep violet **PyTPA** solid absorbs at λ = 304, 403, and 660 nm, where 660 nm absorption was maximum (Fig. S1b). The **PyTPA@WP**, was treated with different amines ($10^{-5}$ M solution in DMAc) and also amine vapors (wherever possible). The respective changes are presented in Fig. 8a, b. Although the color change was slightly different compared to the solid state due to the trace amount of solvent, the sensitivity of the selective amines was very similar in both phases. Thus, **PyTPA@WP** can monitor selective amine vapor/solution expediently and directly.

**Quantitation in the vapor phase**
**PyTPA@WP** were exposed to various amine vapors to monitor the apparent color change of the probe under ambient and 365 nm UV light. The above results on rapid color change motivated us to investigate concentration-based color variation upon vapor exposure. Glass jars (Fig. 8e) were made ready to screen PUT and CAD liquid (vapor pressure at

298 K (mm Hg): 2.3 for PUT and 0.97 for CAD)[52]. Notably, PUT and CAD vapor are the most crucial indicators of food spoilage. These amines were added through the septum in different µL (1–50 µL) volumes using a Hamiltonian micro syringe, and the jar was kept at 298 K. The respective absorbance at 660 nm of **PyTPA**-casted paper strip was monitored upon gradual increase with CAD and PUT concentration (Fig. 8c, d), displaying a prompt reduction of absorption. Additionally, there was a visually measurable slight disappearance of blue coloration at 4.35 mg/L, and it gradually faded away and became complete upon reaching 43.5 mg/L. After reaching the equilibrium, the color change remains intact even at higher concentrations. Thus, we could conveniently and unequivocally recognize the color difference upon reaching the CAD and PUT vapor concentration of 21.75 mg/L (Fig. 8f, g).

**Quantification of CAD (solution) using PyTPA on the paper strip**
Further, we examined the **PyTPA@WP** by dropping CAD solution at different concentrations starting from $10^{-2}$ M to $10^{-9}$ M in DMAC as solvent. Visual detection at the level of $10^{-7}$ M can be reached, as presented in Fig. 8h.

**Quantification of CAD (vapor) using PyTPA on the glass coverslip**
Furthermore, the **PyTPA**-coated glass coverslip was exposed to CAD vapors to monitor the apparent color change of the probe under normal light. A slight disappearance of deep violet coloration at 32.62 mg/L (Fig. 9) was noticed, and it gradually turned to a sort of orange coloration completely upon reaching 54.37 mg/L. Therefore, the color change remains intact even at higher concentrations of CAD. However, the response time was higher

**Fig. 8 | Colorimetric responses on paper strips.** Color change of **PyTPA** in **a** solid (probe) (10 µl of 10 µM drop cast on Whatman filter paper) -solution (analyte) (1 µM), **b** solid (probe) (10 µl of 10 µM drop cast on Whatman filter paper)-vapor (analyte) for various amine in ambient light and under 365 nm UV light. Where; (i) **PyTPA** (10 µM), (ii) control, (iii) 1,3-DAP, (iv) CAD, (v) PUT, (vi) 2-PEA, (vii) nicotine, (viii) spermine, (ix). spermidine, (x). ammonia, (xi). TEA. **c** UV-Vis spectra for solid-vapor phase titration of **PyTPA** with **c** CAD and **d** PUT vapor upon increasing vapor density of amine vapors, **e** Setup image used for vapor phase titrations. Photograph of **PyTPA** paper strips in ambient light upon exposure with **f** CAD and **g** PUT vapor, **h** quantification of CAD (solution) using **PYTPA@WP**.

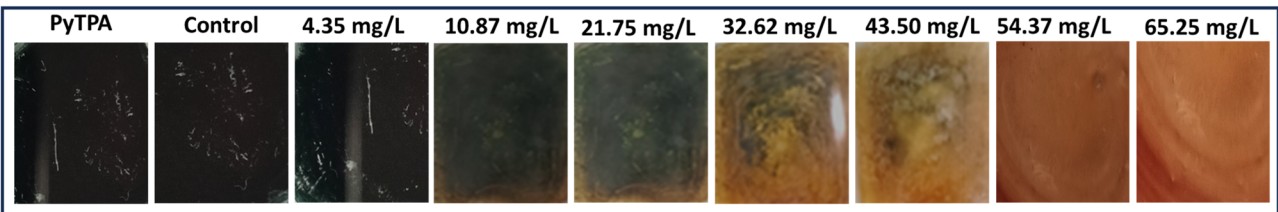

**Fig. 9 | Colorimetric responses on a glass surface.** Ambient light color variation of PyTPA glass with different concentrations of CAD vapor.

(6 h) compared to the paper strip (30 min). Typically, a higher concentration of **PyTPA** ($10^{-2}$ M **PyTPA** in 1–4 dioxane) was needed to make the film over the glass coverslip through the drop-casting method. Therefore, a satisfactory response can only be obtained from such a glass platform if the vapor concentration is adequate. Thus, paper strip would be a better choice in terms of sensitivity as it can display the responses even from 4.35 mg/L concentration. to 32.62 mg/L.

### Mechanistic insights and experimental support

The fundamental reason behind this absorption reduction is the electrostatic interactions between biologically relevant amines and **PyTPA**, followed by a conversion from pyrylium to pyridinium ion (Fig. 10a), a well-

recognized fact due to the higher reactivity of pyrylium ion. The combined effect of diamine interactions with **PyTPA** and chemical reactions make the system highly sensitive up to *nM* level detection. However, styrylpyrylium salts are still unfamiliar in colorimetric detection, so we attempted to find the cause behind such photophysical changes in both the solution and solid states. First, Job's plot (Fig. S11a–e) indicated a ~ 1:2 (analyte: probe) binding for the biogenic amines (longer chain length with terminal -NH$_2$) such as 1,5-DAP (CAD) and spermine, whereas ~1:1 binding was followed for 1,3-DAP (shorter chain length diamine), 2-PEA, and histamine (holding a single NH$_2$). The same was evidenced by mass-spectrometric studies. The mechanism of this salt formation is very well recognized in the literature, mainly with monoamines, *not diamines*. However, formation of

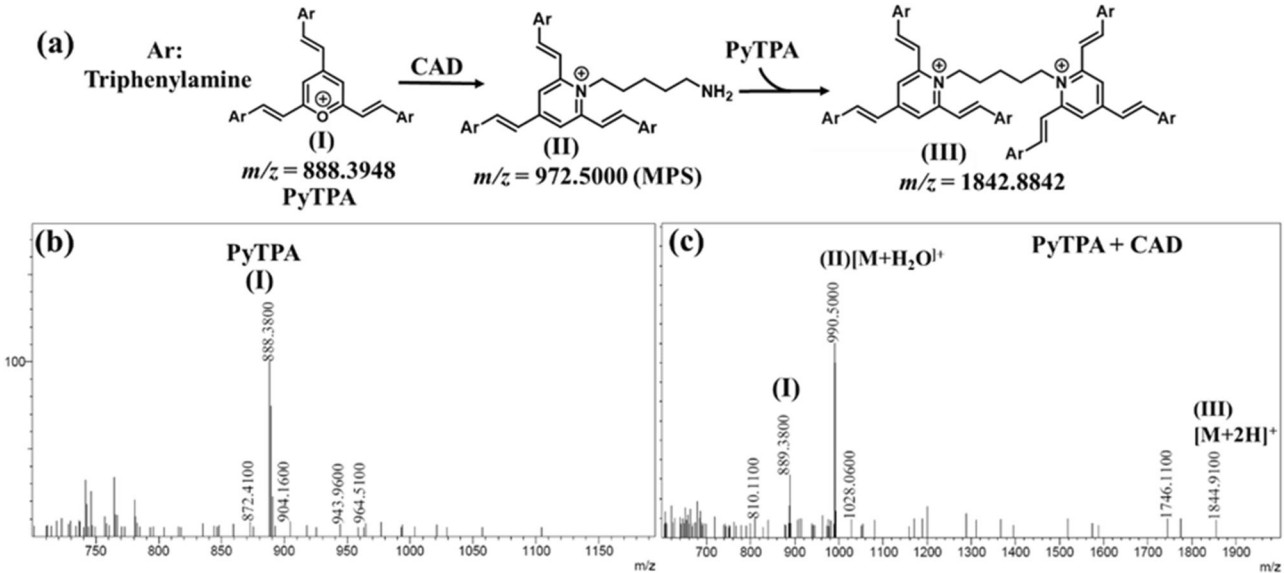

**Fig. 10 | Generation of pyridinium salts and evidence. a** Identification of mono and di-pyridinium salts from the reactions between PyTPA with CAD; the mass spectrum of **b** PyTPA, **c** PyTPA + CAD.

such bis-ammonium complex was stated in pyrylium-based detection of the peptides bearing two terminal amines[53]. However, we could identify the mass ($m/z$) signals corresponding to both mono-and bis-pyridinium salts (Fig. 10b) after the addition of CAD in **PyTPA** (**I**, Fig. 10c) (10 μM in MeCN). Moreover, mono-pyridinium complex (**II**) was observed at $m/z$ of 990.5000 $[M + H_2O-H]^+$ along with bis-pyridinium salt (**III**) at $m/z$ of 1844.9100 [M]. (Fig. 10c). Interestingly, the corresponding mass signals for mono-pyridium salt of histamine $m/z$ 981.4600 for $[M-H]^+$ (Fig. S12a)] and guanidine $m/z$ 931.4200 $[M + 2H]^+$ (Fig. S12b) inferred the formation of mono-pyridinium salt in case of guanidine and histamine. The Job's plot and the mass analysis indicate that the diamines with smaller carbon chains, such as 1,3-DAP and PUT can form only mono-pyridinium salt, whereas CAD and others holding relatively larger carbon chains can also form bis-pyridinium because bis-pyridinium salt needs well-separated Pyridium ions (avoid cation-cation repulsion) and also bulky TPA units (adjust the steric hindrance). This observation was further supported by recording the mass ($m/z$) of the corresponding products (Fig. S12c–e) generated after treating **PyTPA** with 1,3-DAP and PUT (shorter chain length) and spermidine (longer chain length).

Further, the upfield shift of all the [1]H-NMR signals of the **PyTPA** solution (Fig. S13) upon adding CAD indicated the heteroatom exchange from higher electronegative oxygen to relatively lower electronegative nitrogen. Moreover, the upfield[54] signals in the pyridinium salt overlapped and could not be well resolved.

Next, the solid-state detection of the amines was supported with FT-IR. The styryl system develops the conjugations to bring $C = O^+$ stretching frequency ($cm^{-1}$) as low as 1544 (typically for arylated pyrylium, it appears at 1607)[11], which shifted to 1589 due to the formation of $C = N^+$. Stronger electronic conjugation in styryl-linked $C = O^+$ compared to analogous $C = N^+$ is primarily responsible for such a longer frequency shift. The stretching shift near 1450 and 1100 also justifies the difference in aromatic $C = C$ stretching in the pyridinium salt (Fig. 11a), indicating deviations in non-covalent interactions between the aromatic rings. The broadness from sharp diffraction signals in PXRD profiles indicates crystallinity destruction upon amine vapor exposure (Fig. 11b).

Also, the changes in supramolecular interactions between pyrylium and pyridinium salts are also captured in scanning electron microscopic images showing a deviation from a slate-like irregularly shaped **PyTPA** to discrete bead-like morphology with 400–600 nm diameter (Fig. 11c, d). Such a nanosphere formation specifies significant molecular structural change possessing a relatively longer alkyl chain after adding the aliphatic

biogenic amines such as cadaverine. Furthermore, X-ray photoelectron spectroscopy (XPS) showed all the required elements with specific binding energies (eV) (Fig. S14a–h). Notably, the N-atom binding energy of **PyTPA** due to the TPA unit appeared at 399.7 eV, and after the amine treatment, along with the pristine, another signature at 402.2 eV appeared, signifying $C = N^+$ (Fig. S14d, g). Thus, the pyridinium ion formation from pyrylium is very well realized, as shown by numerous experimental proofs.

The selectivity issues mainly hinge on the induction of steric hindrance from hydrophobic nine phenyl groups and accessible cavities in **PyTPA**. The amine selectivity is directed by various factors, including nucleophilicity, steric hindrance, vapor pressure, and probe affinity towards amine. The relatively weaker reactivity and more hydrophilic nature of ammonia (lack of affinity) may make it difficult to interact with this tristyrylpyrylium. The other amines, such as TEA or DIPEA, are relatively bulkier in reaching the electrophilic centers. The flexible aliphatic biogenic amines would have more scope to react due to the multiple possible conformations with two reactive amine centers separated by $CH_2$ units. Once it reacts with one pyrylium unit, the movement of such biogenic amines will be restricted and rejoin with another unit of pyrylium to form the bispyridinium salt separated by $CH_2$ units (as proved through Mass/NMR). Multiple hetero atoms (in the case of histamine, serotonin, spermine, spermidine) and aromatic rings (2-PEA) also create numerous electrostatic interactions to construct supramolecular assembly with **PyTPA**, facilitating the chemical reactions to detect neurotransmitters even with a single –$NH_2$ group. The reactivity of the **PyTPA** was weaker with nicotine as there was no free-$NH_2$ available to react. A temperature of 313 K was needed to activate nicotine to interact with the TPA unit, forming an emissive complex identifiable in mass at 1049.5 for $[M-1]^+$ (Fig. S15). Strong base guanidine reacted similarly to a mono-amine with –$NH_2$, and the $m/z$ of the expected product was identified in the mass (Fig. S12b).

The electrophilic carbons in the pyrylium core gradually interact via electrostatic interactions and respond with monoamine- or diamine-based analytes in 1:1 and 1:0.5 molar ratios, as shown in (Fig. 12a). The positive charge on the oxygen atom makes it suitable for 1,2-addition, not 1,4-addition. The detection of nicotine, which lacks a primary amine group, is likely sensed by host-guest interactions. To understand better, we optimized the ground-state geometry of the fluorophore and the fluorophore-nicotine complex, observing that multiple noncovalent interactions between **PyTPA** and nicotine play a crucial role in the detection. Notably, nitrogen atoms in nicotine are not involved in any reactions or interactions with **PyTPA**. However, olefinic hydrogens take part in forming H⋯H-type close

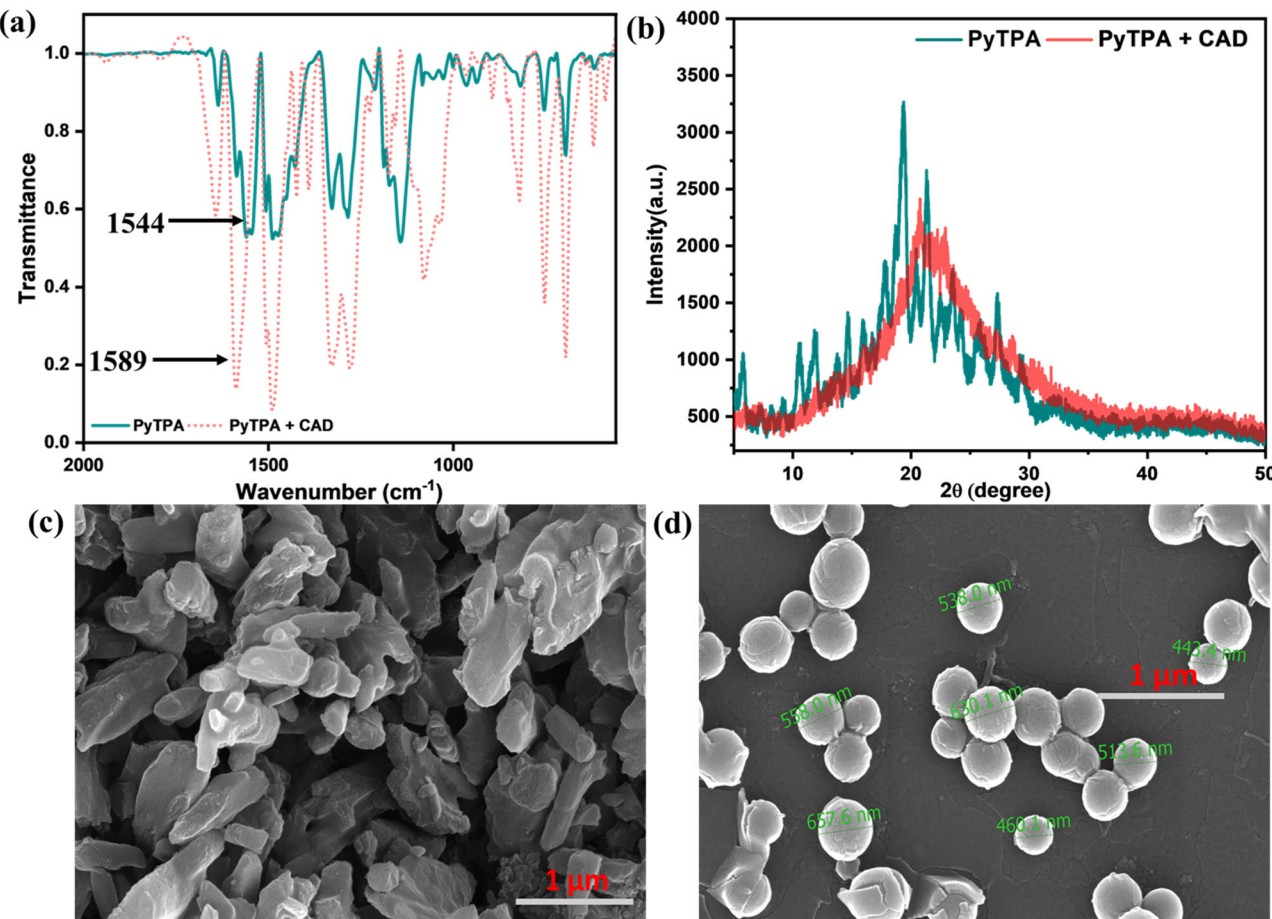

**Fig. 11 | Characterization of solid PyTPA before and after fuming cadaverine vapor.** Comparative **a** IR, **b** PXRD, **c** SEM (before), and **d** SEM (after) analysis before and after fuming cadaverine vapor in **PyTPA**.

interactions (Fig. 12b) with the H atoms of nicotine, with bond distances of $d_2 = 1.925$ Å and $d_3 = 1.868$ Å. The positively charged oxygen in the pyrylium core forms an $O^+ \cdots H$ interaction at a distance of $d_1 = 3.095$ Å. The interaction energy $[E_{int} = E_{PyTPA\text{-}nicotine} - (E_{PyTPA} + E_{nicotine})]$ was calculated using the ωb97xd/6-31+g(d,p) basis set and found to be −28 kcal/mol. This negative value signifies strong dimeric interactions between the probe and analytes.

Measurable emission enhancement only for guanidine can be elucidated with the formed pyridinium salt. The structurally rigid nicotine was revealed to form a supramolecular complex with **PyTPA** at 313 K, which restricts the movement of the probe, promoting excited state relaxation through a radiative path. The strong basic character of guanidine quickly forms the emissive pyridinium salt (as identified in mass, Fig. S12b), possessing no alkyl chains, unlike other biogenic amines. Such pyridinium salts with $NH/NH_2$ functionalities would possibly create supramolecular interactions with the styryl part of **PyTPA**, resulting in a conformationally rigid complex displaying emission. This explanation also relates to emission enhancement observed for almost all the biologically relevant amines. Further, their emission features were supported by the excited state lifetime studies to find the difference in the radiative ($k_r$) and nonradiative ($k_{nr}$) rate constants (ns$^{-1}$) for the probe after the addition of CAD, guanidine, and nicotine. The order of comparative $k_r$ values follows guanidine > nicotine > CAD and agrees with the experimental outcomes (Fig. S16, Tables S3, S4).

**Real-world applications**

The rapid response of **Py-TPA** against various amines prompted us to adopt this technique in everyday life. In this context, the spoilage of the chicken or fish was monitored with the developed probe **PyTPA**, which changed its color against a control exposed to only moisture. The protein-rich food items (here meat/fish) release a few biogenic amines through decarboxylation of amino acids that primarily include spermine, spermidine, putrescine, cadaverine, tryptamine, phenylethylamine, histamine, and tyramine. Out of these amines, aliphatic amines will have a much better vapor pressure at room temperature and thus come into contact with the dye that can detect such vapors by gradually fading its blue color. For the spoiled food, we will have adequate microorganisms present that will make the decarboxylation process much faster, and thus, the amount of released biogenic amines will also be more considerable in concentration. In our experiments (stated above), we noticed that the color change starts from 10 mg/L and responded very well with a complete conversion from blue to yellowish above 50 mg/L. Thus, we can monitor the food's freshness. If the color changes immediately, it will indicate that the food is spoiled, and thus, precautions need to be taken before consumption. If we see that the color starts fading away slowly, we should make proper planning (adequate storing, maintaining temperature, avoiding exposure to microbes, etc.) to avoid the decarboxylation process. Thus, we can prevent food spoilage and also avoid food wastage.

We have attempted to demonstrate the response with some real samples. A small piece of fresh chicken (Fig. 13a) and tuna fish (Fig. 13b) were taken in a small 200 mL closed jar, and the probe was hung freely *via* a paper pin. The color change was gradually monitored under daylight. As these chicken and fish items are in less quantity, the needed duration would be more to release the corresponding biogenic amines vapor upon spoilage and reach an adequate quantity for detection. A slight change (reduction in the color intensity) was detected after 12 h. Tuna fish was chosen for this experiment because of its worldwide use and higher biogenic amine content, including histamine in spoilage. However, distinguishable changes were

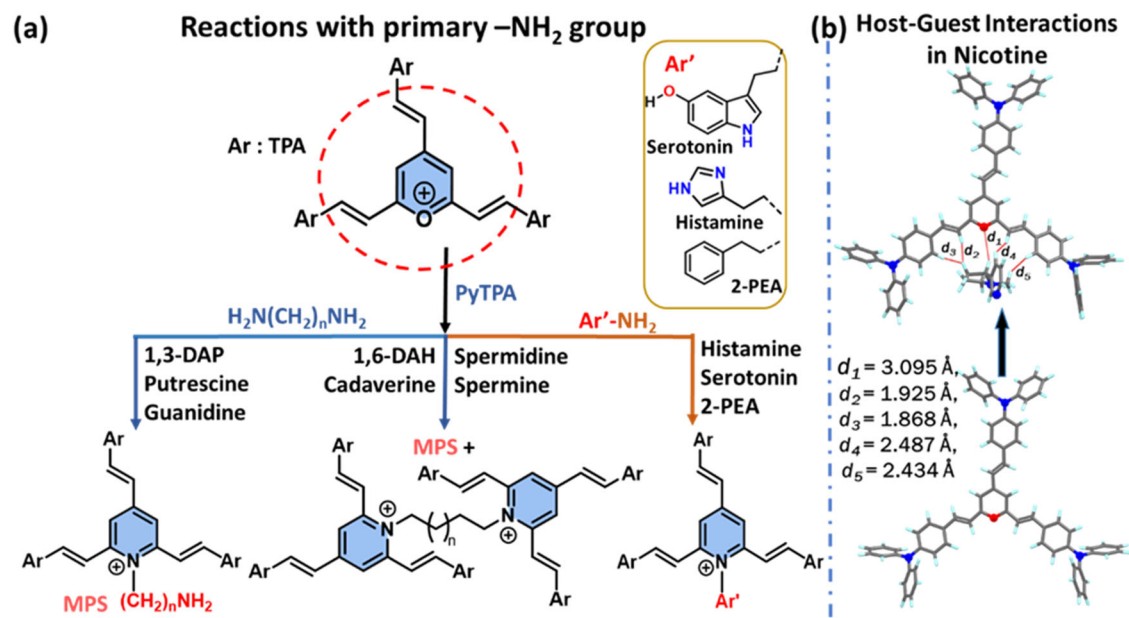

**Fig. 12 | Mechanistic variations to detect various amines.** **a** for aliphatic and aromatic amines bearing –NH$_2$ group, and **b** Nicotine sensing *via* host-guest interactions.

**Fig. 13 | Detection of fish and chicken.** Application of the probe in detecting the freshness of **a** chicken, **b** tuna fish.

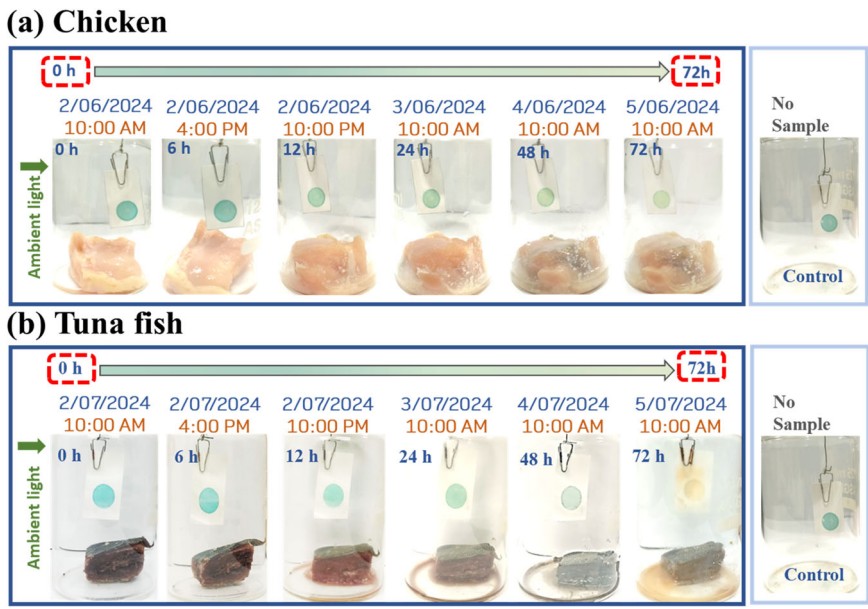

noticed after 12 h, possibly because of amines with higher vapor pressure. Thus, we could monitor the spoilage of different protein-rich foods under normal light using this colorimetric probe.

Furthermore, nicotine detection in the natural sources was conducted in tobacco leaves (TL) and nicotiana, and the cross experiment was performed with another non-nicotine-containing leaf, Caesalpinia Sapan (CSL) (Fig. 14a, b). The leaves were added to 10$^{-5}$ M **PyTPA** in MeCN and heated at 313 K for 30 min. We noted an ambient color change along with an emission shift. Notably, only TL in MeCN was emissive, gradually changing its emission upon increasing quantities of TL. Such a change in emission was not identified with CSL (leaves with no nicotine) (Fig. 14d). The blue color of **PyTPA** started fading away with the gradual addition of TL (Fig. 14c). However, the color remained intact for other CSL leaves.

## Conclusion

A triphenylamine-linked tristyrylpyrylium salt is introduced as an atypical single dye for multiphase detection of crucial biological amines. Contrary to the previous system, this structurally and sterically unique design with multiple hydrophobic units does not allow ammonia vapor to respond, but a promising response was found with diverse biologically relevant amines. Notably, not only aliphatic biogenic amines, the response was equally effective for aromatic biogenic amines such as histamine and serotonin. Most of these responses were detectable by absorbance reduction and thus visible under ambient light. Unlike all the amines, the other two amines, such as guanidine and nicotine, could affect both the absorbance and emission profile, easily recognizable through the naked eye. The cause of these photophysical responses is primarily attributed to the probe's affinity

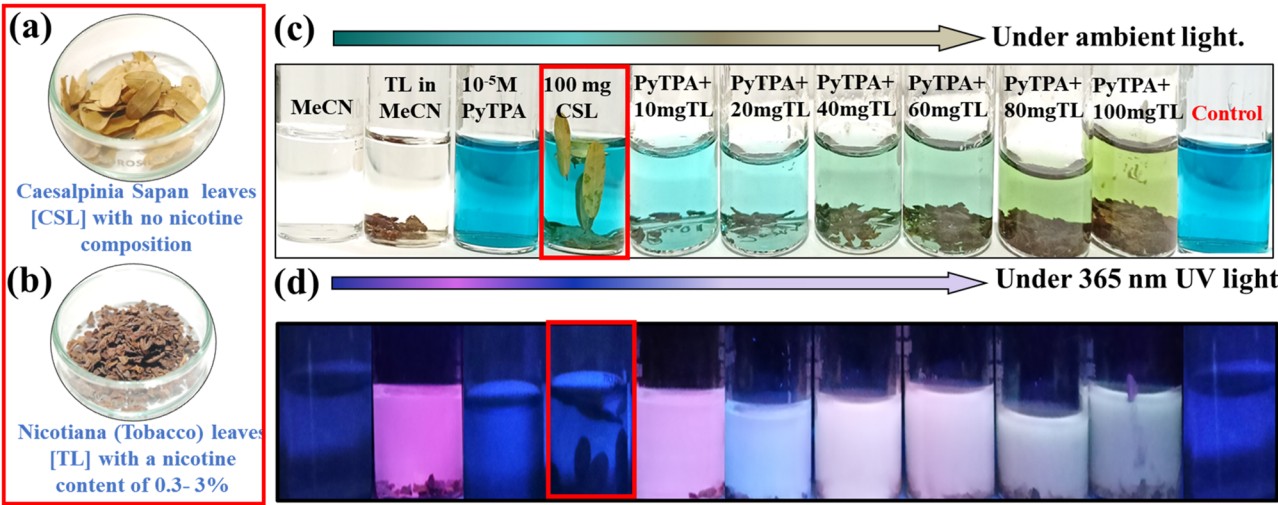

**Fig. 14 | Nicotine detection in solution with real-life samples. a** Caesalpinia Sapan leaves [CSL] **b** Nicotiana (Tobacco) leaves [TL], **c** Photograph of **PyTPA** solution (10 μM in MeCN) upon increasing TL leaves and corresponding change under ambient light, **d** Photograph of **PyTPA** solution (10 μM in MECN) upon increasing TL leaves and corresponding change under 365 nm UV light.

with amines through electrostatic attraction and gradual formation of pyridinium salts, substantiated by various experimental and theoretical evidence. Dissimilar mechanistic paths are identified with different amines. Finally, we could demonstrate the onsite applications to test the freshness of chicken and fish. The dye is used as the ink of an easy-to-carry pen to develop a much easier solution to find the presence of the biologically relevant amines after reaching the *nM* range concentration. The present study significantly contributes to pyrylium chemistry with a potential design suitable for detecting diverse biologically relevant amines, extended to electronics-free real-world applications.

## Methods
### General experimental information
All the required chemicals were purchased from various companies and used without purification. All the products were characterized by [1]H, and [13]C NMR spectroscopy. NMR spectra were recorded on a Bruker 400 MHz instrument (400 MHz for [1]H NMR, and 101 MHz for [13]C NMR). Copies of [1]H,[13]C, and [31]P NMR spectra can be found at the end of the supporting information. [1]H NMR experiments are reported in units, parts per million (ppm), and were measured relative to residual DMSO (2.50 ppm) in the deuterated solvent. [13]C NMR spectra are reported in ppm relative to deuteron DMSO (39.52 ppm) and all were obtained with [1]H decoupling. Coupling constants were reported in Hz. Reactions were monitored by thin-layer chromatography (TLC). liquid chromatography-mass spectrometry (LC-MS) was obtained by the electron spray ionization (ESI) technique using the Q-TOF mass analyzer and was reported as *m/z* (relative intensity). Melting points of compounds were recorded on a KRUSS Optronic M3000 apparatus.

### General experimental procedure for 2,4,6-tris((*E*)-4-(diphenylamino)styryl) pyryllium
An oven-dried round bottom flask was equipped with a magnetic stir bar and charged with 2,4,6- trimethylpyrylium tetrafluoroborate (TMP-BF₄) (42.00 mg, 0.2 mmol), triphenylamine aldehyde (327.73 mg, 1.2 mmol) and acetic acid 10 mL under nitrogen. Then the mixture was heated to 120 °C and stirred for 72 h in nitrogen atmosphere. After cooling to room temperature, the solvent was evaporated to afford a crude residue. The crude product was purified by washing with diethyl ether, methanol, distilled water, and finally with n-pentane. The residue was then dried in a vacuum oven at 60 °C for 1 h.

### Characterization of PyTPA
Shiny deep violet powder (330.70 mg, 85%), m.p.: 196–198 °C, IR (v cm⁻¹) = 3054, 1638, 1586, 1561, 1544 (C = O⁺), 1510, 1488, 1330, 1288, 1143. ¹H NMR (400 MHz, DMSO-d⁶): δ 8.13 (d, ³J_{H-H} = 15.6 Hz, -C=CH_a, 1H), 8.03 (d, ³J_{H-H} = 16 Hz, -C=CH_b, 2H), 7.81 (s, Py-*H*, 2H), 7.72 (d, ³J_{H-H} = 8.8 Hz, 4H, Ar-*H*), 7.69 (d, ³J_{H-H} = 8.8 Hz, 2H, Ar-*H*), 7.45–7.39 (m, 11H, Ar-*H*), 7.26–7.16 (m, 22 H, Ar-*H*), 6.94–6.89 (m, 6H, Ar-*H* + -C = C*H*); ¹³C NMR (101 MHz, DMSO-d⁶): δ 166.4, 151.8, 151.0, 146.2, 145.9, 142.8, 132.0, 131.2, 130.5, 127.4, 126.8, 126.4, 126.0, 125.6, 120.1, 120.0, 119.6, 115.7, 114.3 (four quaternary carbons gave poor signals to identify); HRMS (m/z): [M]⁺ calcd. for $C_{65}H_{50}N_3O$, 888.3948; found, 888.3963.

### Measurements and instrumentations
**Thermal analysis study**. Thermogravimetric analysis (TGA) was conducted on a Shimadzu DTG-60 simultaneous DTA-TG apparatus with an increasing temperature rate at 5 °C min⁻¹ in the N₂ atmosphere.

*Steady-State Absorption and Fluorescence Measurements*. The electronic absorption spectra in the solution state were recorded with a UV 3600 Plus (Shimadzu). The fluorescence spectra were recorded on a Hitachi spectrofluorometer (F7000, Japan) using a 1 cm path-length quartz cuvette. A stock solution of 10⁻³ M was prepared in DMAc, and for histamine, guanidine, and serotonin, stock solution of 10⁻³ M was prepared in Water. The final concentration of the **PyTPA** was adjusted to 10 μM.

**Preparation of amine solutions and titration**. The amines were solubilized in DMAc. histamine, guanidine, and serotonin were solubilized in water and were maintained at (pH- 8–9) using 0.1 N NaOH solution. Further, the absorption and emission spectra were recorded for **PyTPA** (10 μM) followed by adding different fractions (volumes) of amines from a stock solution of 10⁻⁴ M.

*Detection Limit Calculation*. A series of absorbance titrations was carried out to detect amines using **PyTPA** (10 μM in MeCN). The detection limit was calculated by LOD = 3.3 σ/K, where σ is the standard error, and K is the slope of the linear fit.

**Paper strip quenching study(vapor)**. **PyTPA@WP** was exposed to different amine vapors for 10–15 min. The pictures were captured in ambient light under and 365 nm UV lamp using a cell phone camera.

*Paper Strip Quenching Study(solution)*. **PyTPA@WP**, was treated with different amine solutions (10⁻³ M) and was dropped via micropipette over

**PyTPA@WP**. Additionally, for titration, a similar method was employed with different concentrations of amine ($10^{-2}$ M to $10^{-9}$ M), and the pictures were captured in ambient light under a 365 nm UV lamp using a cell phone camera.

**Quantitation using paper strips(vapor)**. **PyTPA@WP** was freely hung inside the glass jar (200 ml) via a paper pin tied to thread and was maintained at the center of the glass jar without touching the wall and was closed tightly with a silicon septum, further properly enclosed with Teflon tape. further calculated amount of cadaverine and putrescine was added via Hamilton syringe through silicon septum. The color change was monitored and was removed after 30 min, and absorbance was recorded.

*Lifetime Decay Measurement.* Time-resolved fluorescence measurements were performed using a time-correlated single-photon counting (TCSPC) unit (Horiba Deltaflex). The laser used for all samples was 403 nm. All measurements were performed at room temperature. The decay fitting was completed, keeping the $\chi^2$ value close to unity.

**Powder X-ray diffraction measurement**. PXRD measurements were carried out using a Xenocs Nano-in Xider SW-L SAXS/WAXS 181 system with a dual detector with Cu Kα micro focus within the range of 5° to 50° at a scanning 182 speed of 2 °/min. The sample was placed and spread over a Kapton tape, and data were recorded 183 in transmission geometry.

*Field-Emission Scanning Electron Microscopic Study (FE-SEM):* The FE-SEM images were carried out for solid dispersed in carbon tape using the FEI Apreo LoVac instrument.

XPS *Studies*: were performed with the help of a Thermo Scientific K-Alpha surface-analysis spectrometer housing Al $K_\alpha$ as the X-ray source (1486.6 eV). The base pressure at the analyzing chamber was maintained at $5 \times 10^{-9}$ mbar. The data profiles were subjected to a nonlinear least-squares curve-fitting program with a Gaussian–Lorentzian production function and processed with Avantage software. The binding energy (B.E.) of all XPS data was calibrated versus the standard C 1 s peak at 284.85 eV.

**Infrared spectroscopic studies**. The IR spectra were recorded using an FT-IR spectrometer KBr-mixed was used as a matrix for recording the FT-IR/ATR-FT-IR spectra.

## Data availability
All data generated or analysed during this study are included in this published article (and its supplementary information files).

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

## Acknowledgements

We thank LSRB [389/FSH&ABB/2021] and DST-PURSE/SATHI for support. The instrumental facility by DST-FIST [SR/FST/CS-I/2020/158] is acknowledged. S.T. and B.P. thank BITS-Pilani Hyderabad Campus for their fellowship.

## Author contributions

S.T.: Synthesis, characterization, photophysical studies, Data preparation, manuscript reviewing; B.P.: Photophysical studies, theoretical studies; Manuscript preparation; M.C.: Planning of the overall work, Supervision of the synthesis, interpretation of the overall result, Manuscript writing and reviewing. All authors deliberated on the results and commented on the manuscript.

## Funding

## Competing interests

The authors declare no competing interests.
