## [Peer review file · Communications Chemistry]

Multiphase Detection of Crucial Biological Amines using Single 2,4,6-Tristyrylpyrylium Dye

Corresponding Author: Professor Manab Chakravarty

Version 0:

Reviewer comments:

Reviewer #1

(Remarks to the Author)

This manuscript presents a comprehensive study on the synthesis and characterization of a pyrylium dye and his detection ability towards biogenic amines, nicotine, and guanidine.

Although the study is of some interest, it requires some major revisions to be suitable for publication.

-The Introduction is too long-winded, it should be shortened and better focused on the aim of this study. This also applies in part to the Abstract.

-The dye does not allow to "identify various aliphatic/aromatic BAs" because its optical absorption response in solution or deposited in the filter paper is almost the same for the investigated diamines. A different behavior (in fluorescence) is observed for nicotine, and guanidine. Therefore, the manuscript should be revised accordingly.

- No information is reported on the optical absorption response of the dye, either in solution or in vapor phase, for aliphatic primary or secondary amines. This would enforce the topic of the paper on "Detection of Crucial Biological Amines", otherwise it would represent just one of the many papers reported for detection of amines and, therefore, of little interest to a wide readership.

Reviewer #2

(Remarks to the Author)

This paper has investigated the response and mechanism of 2,4,6-Tristyrylpyrylium dye for the detection of amines.. It is well written and includes relevant information related with the reactions and possible applications dye-amines. In my opinion it should be published after minor revisions:

- Authors evaluate LOD in solution but not in the solid state of the probe (solid-liquid) or in vapor phase. Please include those data in the text.

- In solution authors evaluate the response of 1,4 DAB and 1,5 DAB. They should use PUT and CAD respectively. Because it is important in the text to compare the same amines with liquid and solid Dye.

- Author should include in experimental section how to calculate K for LOD equation.

- You are calculating K (slope) in a range of values with a blank and 10 M (Fig 6), but which was the minimum concentration of the calibration?

- Authors include along the text (i.e. page 3), some comments about selectivity, however, did you study the effect of different compounds in these results? It is possible to mix different amines?

- This dye is water soluble, why do not you assay this work in water solutions?

Reviewer #3

(Remarks to the Author)

Recommendation: Make minor revisions as directed..

Comments:

The manuscript introduces a 2,4,6-tristyrylpyrylium dye probe for multiphase detection of biogenic amines. The probe demonstrates diverse response mechanisms to biogenic amines in different phases. While the study offers interesting findings, with potential applications such as food spoilage detection and nicotine identification in tobacco, the overall innovation and practical relevance could benefit from further exploration. The methodology is sound, and the results are

noteworthy, but some areas require improvement. I recommend considering the manuscript for publication after revisions.

1. How does PyTPA perform in amine detection at different temperatures? For example, can it maintain reactivity with amines at lower temperatures, or does it require heating, as seen in nicotine detection?
2. What are the advantages and disadvantages of PyTPA compared to other amine-detecting dyes? In terms of sensitivity, stability, and response time, does it outperform existing dyes?
3. In a complex mixture environment, does PyTPA also react with non-amine molecules? How can the high selectivity of PyTPA for amines be ensured?
4. The references cited in the manuscript are somewhat outdated. It is recommended to include more recent studies to ensure the manuscript's relevance in the current academic context. For example, refer to the recent publication Chinese Chemical Letters, 2024, 35(3), 108579, Sensors and Actuators B: Chemical, 2024, 409, 135563 to enhance the timeliness of the literature review and further support the discussion of the research background.

Reviewer #4

(Remarks to the Author)

The paper covers pyrylium salts and the use of one specific organic material for amine detection. The topic is relevant and of interest to a general chemistry audience. The reported chemical structure is new and worthy of being reported. But I found the quality of writing to be low and there is a lack of quantitative data. I would suggest the authors resubmit a revised paper.

Writing/grammar can be improved. I found the paper very hard to read. Please proofread.

Careful with the hyperbole. For example, line 31, why are amines, carbonates "crucial". Please make specific statements. Why is biogenic amine abbreviated to BA? I find this confusing. Just write out biogenic amine.

NMR spectra must be assigned.

Double check coupling constants. Type not listed. What is coupling to what? Constants must match.

NMR must be followed by spectra, spectrum, etc.

Figure 1 is quite crowded and confusing. Looks more like a "poster presentation" or "graphic table of contents" than a paper Figure. Please adjust.

Not clear on the hypothesis for using the triphenylamine. This unit is an electron donor and thus should increase the internal charge transfer characteristic of the molecule. How would this impact the O for N substitution? What is 'decent solubility'. Please quantify and correlated to what is required for the application.

The authors state the TPA has a high hole mobility? Where is the data? In fact all text on lines 105-110 need back up data. Is a thermal stability of 250oC good? Please compare and contrast to related compounds.

Why test the photostability over only 2 days? What is normal light? The light intensity must be measured and reported. Use a continuous illumination source. Otherwise not relevant.

For the solid-state experiments please characterize the material on the filter paper. What is the nature of this material? Why not solution coat onto glass to create films? Reason for this method should be added.

The detection limits should be tabulated. How do they compare to the literature. What are the required detection limits for real-time use?

I do not follow the section "real-world applications". The sensor is detecting amines emitted from the decaying meat, but what does this mean? How does one know if the food is spoiled or not? We know that the food is going bad, so what added value does the sensor give us? I do not see any reason for this sensor unless the color change can be correlated to the quality of the product. The nicotine detection seems more appropriate as it gives a yes/no answer. The pen section (Figure 13) is not scientific. If biogenic amines are present, it does not mean the food is bad. Seems this would lead to more food waste due to false information. This entire section should be reworked or removed. If reworked include industry standards.

Version 2:

Reviewer comments:

Reviewer #1

(Remarks to the Author)

The revised manuscript satisfactorily addressed all my comments.

Reviewer #3

(Remarks to the Author)

Thank you to the authors for their revisions and responses to the reviewer comments. The revised manuscript adequately addresses all the issues raised, enhancing the comprehensiveness of the experimental data and the timeliness of the literature. Below are the specific comments:

1. Temperature performance: The experimental results show that PyTPA performs well at different temperatures.
2. Comparison of advantages and disadvantages: The authors have provided a detailed list of the advantages and limitations of PyTPA, and a comparative table clearly demonstrates its superiority over other dyes.
3. Reaction with non-amine molecules: The authors conducted further selectivity experiments, and the results show that PyTPA does not react with non-amine substances, validating its excellent selectivity.
4. Literature update: The authors have updated the references, incorporating recent relevant studies, which strengthens the

timeliness of the literature review.

Overall, the revised manuscript satisfactorily addresses the reviewer comments, with significant improvements. I recommend the manuscript be accepted for publication.

Reviewer #4

(Remarks to the Author)

The authors had addressed all comments. While there is still work to do, the papers seems OK to be published at this time. I would state the work is of scientific value but is more incremental than innovative considering there are countless studies on organic amine sensors.

Responses to Reviewers

Reviewer #1 (Remarks to the Author):

This manuscript presents a comprehensive study on the synthesis and characterization of a pyrylium dye and its detection ability towards biogenic amines, nicotine, and guanidine. Although the study is of some interest, it requires some major revisions to be suitable for publication.

Response: Thank you for reviewing our manuscript and providing your valuable thoughts. We highly appreciate your views on this newly developed colorimetric probe. It is our immense pleasure to address your concerns. Moreover, your comments inspired us to think more about improving the manuscript's quality. Thank you so much.

Comment 1: The Introduction is too long-winded, it should be shortened and better focused on the aim of this study. This also applies in part to the Abstract.

Response: Thank you so much for your suggestions. We have modified the introduction and made it relatively shorter. Only important points are stated in the introduction. The abstract is maintained now within 200 words. Hope, it will be reasonable now. Thank.

Comment 2: The dye does not allow to “identify various aliphatic/aromatic BAs” because its optical absorption response in solution or deposited in the filter paper is almost the same for the investigated diamines. A different behavior (in fluorescence) is observed for nicotine, and guanidine. Therefore, the manuscript should be revised accordingly.

Response: Thank you so much for your suggestions. Yes, it cannot identify but can sense some of these specific amines, including aliphatic/aromatic biogenic amines, guanidine at 25 °C and nicotine (at 40 °C). However, the mechanism includes the supramolecular interactions and is followed by the amine addition in 1,2-fashion. Thus, it does not prefer typical monoamines (1°/2°/3°) but chooses selectively diamines (such as BAs) or other heteroatom-possessing monoamines. Even guanidine and nicotine can be detected via colorimetric/fluorimetric routes. We have modified the same in the revised manuscript. Thanks.

Comments 3: No information is reported on the optical absorption response of the dye, either in solution or in vapor phase, for aliphatic primary or secondary amines. This would enforce the topic of the paper on “Detection of Crucial Biological Amines”, otherwise it would represent just one of the many papers reported for detection of amines and, therefore, of little interest to a wide readership.

Response: Thank you so much for your input. Now, we have tested many other amines, alcohol, amino alcohols, and other related analytes. We earlier tested ammonia, but now we verified other amines: primary amines [ethylamine, n-butylamine, benzylamine, aniline], secondary amines [diethylamine(DEA)], tertiary amines [diisopropylethylamine (DIPEA)] and the mixture of tertiary and primary amine i.e. N, N-Dimethyl-1,3-propanediamine (DMPA). We found no change in the absorption for almost all the amines. Only DMPA showed a slight decrease in low energy absorption (Fig 1). This result is included in the revised manuscript.

Fig 1: UV-vis spectra of **PyTPA** (10 μM in MeCN) upon addition of respective primary, secondary aliphatic/aromatic amines (10 μM in DMAc) at 298 K.

Reviewer #2 (Remarks to the Author):

This paper has investigated the response and mechanism of 2,4,6-Tristyrylpyrylium dye for the detection of amines. It is well written and includes relevant information related with the reactions and possible applications dye-amines. In my opinion it should be published after minor revisions:

Response: Thank you for reviewing our manuscript and providing constructive suggestions. We are pleased to address your comments.

Comments 1: Authors evaluate LOD in solution but not in the solid state of the probe (solid-liquid) or in vapor phase. Please include those data in the text.

Response: Thank you so much for raising this point. Yes, we have quantified the analyte (both cadaverine and putrescine) in the following phases (i) both analytes and probe in solution: LOD: nM level (ii) analyte: vapor and probe in solid state (paper strip): It could detect up to 4.35 mg/L level. Now, we also checked the other option as suggested (iii) analyte (cadaverine) in solution and probe in solid (paper strip). We noticed the cyan-greenish paper faded until the concentration of 10^{-7} M (100 nM). We have added this part to the revised manuscript. Thank you so much.

Comments 2: In solution authors evaluate the response of 1,4 DAB and 1,5 DAB. They should use PUT and CAD respectively. Because it is important in the text to compare the same amines with liquid and solid Dye.

Response: Thank you so much for your valuable suggestions. We have replaced 1,4-DAB with PUT and 1,5-DAP with CAD to maintain the uniformity.

Comments 3: Author should include in experimental section how to calculate σ for LOD equation. You are calculating K (slope) in a range of values with a blank and 10 μ M (Fig 6), but which was the minimum concentration of the calibration?

Response: Thank you for your concern about the LOD calculation. We calculated LOD using the equation $LOD = 3.3 \times \sigma/K$

Where σ = Standard error of the linear fitting (highlighted in red in the graph below) and K = Slope

e.g. For **1,3-DAP**, $\sigma = 0.01962$ and $K = 1.22 \times 10^6 M^{-1}$, thus $LOD = \frac{3.3 \times 0.01962}{1.22 \times 10^6} = 53 \times 10^{-9} M$

We have titrated in the range of 0.5-10 μ M of the analyte to the sensor solution. Thus, the minimum concentration of the calibration was 0.5 μ M.

Comments 4: Authors include along the text (i.e. page 3), some comments about selectivity, however, did you study the effect of different compounds in these results? It is possible to mix different amines?

Response: To find the selectivity, we noticed that aliphatic monoamines were nonresponsive. In fact, the combination of amine/alcohol was also not responsive. This result is included in the revised manuscript.

To compare the nucleophilicity between alcoholic and phenolic groups, we performed the sensing using methanol, 1,4-butanol, 4-aminobutanol, ethylene glycol (EG), phenol, catechol and resorsinol, exhibiting only a slight change in the absorption.

Thanks for your suggestions. We have investigated the changes in the absorbance spectra after mixing different amines. At the beginning, upon the addition of 10 nM PUT (in DMAc) to 10 μ M of PYTPA (in MeCN), the absorption at 665 nm reduces (~23% of the initial absorption) even with such a small amount of PUT. Next, 10 nM CAD (in DMAc) also displayed a similar reduction in absorbance. To compare this reduction, 10 nM DIPEA (in DMAc) was added and that revealed only 15% absorption lessening, as observed in the initial screening. Later, 20 nM of sperimidine (DMAc) was treated, and a 77% reduction in absorbance was observed. Thus, this dye could detect important BAs even in the mixture of various amines.

Comments 5: This dye is water soluble, why do not you assay this work in water solutions?

Response: Thanks for your concern about making it a greener approach. It is not water-soluble due to the presence of nine phenyl groups that makes the system very much hydrophobic. The stock solution was dissolved in water, and some particles precipitated from this aqueous solution. We see relatively lesser absorption in water. Therefore we refrain from testing in water. Thanks.

Reviewer #3 (Remarks to the Author):

Recommendation: Make minor revisions as directed..

Comments: The manuscript introduces a 2,4,6-tristyrylpyrylium dye probe for multiphase detection of biogenic amines. The probe demonstrates diverse response mechanisms to biogenic amines in different phases. While the study offers interesting findings, with potential applications such as food spoilage detection and nicotine identification in tobacco, the overall innovation and practical relevance could benefit from further exploration. The methodology is sound, and the results are noteworthy, but some areas require improvement. I recommend considering the manuscript for publication after revisions.

Response: Thank you so much for scrutinizing our manuscript and giving your constructive thoughts. We highly praise your views and are pleased to address the comments. Your comments have assisted us in thinking in a much deeper sense and improving the quality further.

Comment 1: How does PyTPA perform in amine detection at different temperatures? For example, can it maintain reactivity with amines at lower temperatures, or does it require heating, as seen in nicotine detection?

Response: Thank you so much for your suggestions. The experiments are conducted at temperatures -18 °C (255 K), 0 °C (273 K), and 20 °C (293 K). The responses are impressive at all these temperatures. This part is added to the revised manuscript.

Figure: Absorbance spectra of the PyTPA (10 μ M in MeCN) + CAD (10 μ M in DMAc) at three different temperatures.

Comment 2: What are the advantages and disadvantages of PyTPA compared to other amine-detecting dyes? In terms of sensitivity, stability, and response time, does it outperform existing dyes?

Response: A few important unique points on this dye (advantages): (i) It is synthesized using readily available precursors without any chromatographic purification, (ii) It has a decent photo and thermal stability, (iii) The hydrophobic nature of multiple aryl rings does not favor the hydrophilic analytes and thus, offer selectivity. (iv) A single dye to detect only aliphatic, aromatic biogenic amines, nicotine, and guanidine. (v) It can form a nice thin film and get absorbed on paper. (vi) color change is readable through the naked eye under normal light and upon 365 nm UV lamp irradiation. (vii) Multiphase detection through the naked eye under room light/sunlight. (viii) The detection was rapid.

Disadvantages: (i) It needs a relatively high temperature in the synthetic procedure (ii) It is not responsive to catechol amine-based neurotransmitters. (iii) absorbance was marginally reduced after two days of exposure under normal light.

A comparative table is also included in the revised ESI. The uniqueness of this dye is its molecular design and the above-stated advantages, which allow crucial biologically relevant amines to be sensed in various phases. Thus, it outperforms in many ways. Thank you so much.

Comment 3: In a complex mixture environment, does PyTPA also react with non-amine molecules? How can the high selectivity of PyTPA for amines be ensured?

Response: Thanks for such valuable comments that initiated us to examine other non-amine molecules such as alcohols/phenols and amino alcohols. However, the probes detected none of the alcohols/phenols/diols, or even amino alcohols. The absorbance spectra were almost unchanged. The same is added to the revised manuscript. Thanks.

Comment 4. The references cited in the manuscript are somewhat outdated. It is recommended to include more recent studies to ensure the manuscript's relevance in the current academic context. For example, refer to the recent publication Chinese Chemical Letters, 2024, 35(3), 108579, Sensors and Actuators B: Chemical, 2024, 409,135563 to enhance the timeliness of the literature review and further support the discussion of the research background.

Response: Thank you so much for your suggestions. We enjoyed reading these relevant articles. We have cited them in the revised manuscript. Thanks

Reviewer #4 (Remarks to the Author):

The paper covers pyrylium salts and the use of one specific organic material for amine detection. The topic is relevant and of interest to a general chemistry audience. The reported chemical structure is new and worthy of being reported. But I found the quality of writing to be low and there is a lack of quantitative data. I would suggest the authors resubmit a revised paper.

Response: We are very much grateful to you for finding our design interesting and providing valuable thoughts/comments. We have modified the sentences and made them more understandable in the revised version. Thank you so much.

Comment 1. Writing/grammar can be improved. I found the paper very hard to read. Please proofread.

Response: We have processed the draft using Grammarly software and also modified it to improve the overall writing. Thank you so much for your constructive suggestions.

Comment 2. Careful with the hyperbole. For example, line 31, why are amines, carbonates “crucial”. Please make specific statements. Why is biogenic amine abbreviated to BA? I find this confusing. Just write out biogenic amine.

Response: We have reconstructed it specifically, in general analytes. BAs are replaced with biogenic amines as suggested. We just abbreviated it with the thought of saving space, nothing else. Thanks for your suggestions.

Comment 3: Double check coupling constants. Type not listed. What is coupling to what? Constants must match. NMR must be followed by spectra, spectrum, etc.

Response: Thanks for your concern and suggestions. We have checked the NMR spectrum carefully. We could identify a few olefinic protons with trans coupling of ~16 Hz. However, three of them were overlapped with the aromatic protons. As only H's are present for the suitable coupling (^{14}N , is a quadrupolar nuclei with $I=1$), we have mentioned the $^3J_{\text{HH}}$ wherever applicable.

As all the aromatic H's from the triphenylamine part appeared together, no separate coupling could be identified. Thanks.

Comment 4: Figure 1 is quite crowded and confusing. Looks more like a “poster presentation” or “graphic table of contents” than a paper Figure. Please adjust.

Response: It is changed now. It is presented in two different figures to make it clean and more precise. Thank you so much.

Comment 5: Not clear on the hypothesis for using the triphenylamine. This unit is an electron donor and thus should increase the internal charge transfer characteristic of the molecule. How would this impact the O for N substitution? What is ‘decent solubility’. Please quantify and correlated to what is required for the application.

Response: Yes, triphenylamine (TPA) is electron-rich and, thus, increases internal charge transfer. The same is visualized in the absorption spectrum for this molecule. However, TPA has three phenyl rings aligned in a propeller shape, which creates restricted rotation under a suitable environment. Moreover, in this design, three TPA, i.e., nine phenyl rings, boost the hydrophobicity and steric hindrance of the probe, assisting in exhibiting selectivity. The same is explained in the manuscript. Thanks.

The charge transfer is reduced significantly once the O⁺ is substituted with N⁺. The sustainability of O⁺ in the probe is much lower than N⁺ because O is more electronegative than N. However, during this substitution, we noticed fluorescence enhancement for all the probes, specifically nicotine (at 40 °C) and guanidine (at 25 °C).

TPA is slightly soluble in water (<1 mg/ml at 20 °C) but shows very good solubility in organic solvents such as ether, toluene, chlorinated solvent, DMSO, DMF, etc. However, we need very low concentration (<10⁻⁴ mol/L) for spectroscopic studies. Here, we have mentioned the solubility

of TPA. We changed the word “decent” in the revised manuscript. We have modified the sentences. Thanks.

Comment 6: The authors state the TPA has a high hole mobility? Where is the data? In fact all text on lines 105-110 need back up data. Is a thermal stability of 250 °C good? Please compare and contrast to related compounds.

Response: Thanks for your concerns. TPA is known to have high hole mobility (ref: *J. Mater. Chem. A*, 2017,5, 1348-1373). It is primarily important for the semiconductor research. As this biogenic amine sensing work does not directly relate to the hole mobility, we refrain from measuring it for this compound.

Other properties of TPA are very well established. In fact, we could make this aldehyde easily and form a thin layer of it. Now the references are given as back up with the fundamental benefits of triphenyl amine core. *Polymer chemistry* , 2018, 9, 3001-3018 ; Lian, X.; zhao, Z. Cheng D Molecular crystals and liquid crystal, 2017, 648, 223-235.

The thermal stability of organic compounds is a very complex property monitored by various factors. While some compounds are relatively stable, others may decompose or undergo chemical reactions upon heating. This compound with sterically crowded triphenylamine is expected to be thermally stable due to their localized π electrons, sterically crowded molecules hinder its decomposition and hence enhance thermal stability. Comparing other materials and our required applications, 250 °C would be suitable. The thermal stability of the related pyridinium salts has not yet been reported. A few references are included in the revised manuscript to support the statements.

Comment 7: Why test the photostability over only 2 days? What is normal light? The light intensity must be measured and reported. Use a continuous illumination source. Otherwise not relevant. For the solid-state experiments, please characterize the material on the filter paper. What is the

nature of this material? Why not solution coat onto glass to create films? Reason for this method should be added..

Response: Your points are very much praiseworthy. It was tested for 2 days to compare with the related dye **EPY** reported in the literature. Normal light means room light (a mixture of wavelengths). The light intensity is ~ 300 Lux.

Now we have tested the photostability under sunlight (Intensity: 400-10,000 Lux). As we focus on real-world applications, sunlight source is preferred.

We have characterized the material on filter paper through IR, SEM, and UV-Vis spectroscopic techniques.

The probe-coated paper strip was further characterized by SEM, IR, and UV-Vis spectroscopic methods before (PyTPA@WP) and after exposure to amine vapours (PyTPA@WP+CAD) with reference to (WP). It is added to the revised manuscript.

Additionally, we examined the PyTPA-coated glass coverslip upon exposure to CAD vapors to monitor the apparent color change of the probe under day light. We observed a slight disappearance of deep violet coloration at 32.62 mg/L (Figure 8), and it gradually turned orange completely upon reaching 54.37 mg/L. The color change remains intact even at higher concentrations. However, the response time was comparatively higher (6 hours) than in the case of the paper strip (30 minutes). Highly concentrated PyTPA (10⁻² M in 1-4 Dioxan) was drop cast over the cover slip to make a film. Therefore, more amine vapors were needed for the response, and therefore, sensitivity was lower than the paper strip, which helped us to find the response at a very lower limit (4.35 mg/L-paper strip). Hence, paper strip (10 μ M) was superior. Though, paper strip was more effective, and handy

Comment 8: The detection limits should be tabulated. How do they compare to the literature? What are the required detection limits for real-time use?

Response: The detection limit is specified now. It is stated in the Table S6. The real-time user detection limit would be 20mg/L.

Comment 7: I do not follow the section “real-world applications”. The sensor is detecting amines emitted from the decaying meat, but what does this mean? How does one know if the food is spoiled or not? We know that the food is going bad, so what added value does the sensor give us? I do not see any reason for this sensor unless the color change can be correlated to the quality of the product. The nicotine detection seems more appropriate as it gives a yes/no answer.

Response: Thank you so much for your concern. This probe can detect only the freshness of food. If there is a color change, it will indicate that enough BAs are produced, it can sense only after reaching >20 mg/L. It will only provide an indication if the item is eatable or not. If, the color change is prominent, someone should be careful about the food quality.

Comment 8: The pen section (Figure 13) is not scientific. If biogenic amines are present, it does not mean the food is bad. Seems this would lead to more food waste due to false information. This entire section should be reworked or removed. If reworked include industry standards.

Response: Thank you so much for your suggestion. We have deleted this pen section. Thanks

END

REVIEWERS' COMMENTS:

Reviewer #1 (Remarks to the Author): The revised manuscript satisfactorily addressed all my comments.

Response: Thank you so much for reviewing our manuscript and providing your valuable thoughts. We highly appreciate your views. It is our immense pleasure to address your concerns. Moreover, your comments inspired us to think more about improving the manuscript's quality. Thank you so much.

Reviewer #3 (Remarks to the Author):

Thank you to the authors for their revisions and responses to the reviewer comments. The revised manuscript adequately addresses all the issues raised, enhancing the comprehensiveness of the experimental data and the timeliness of the literature. Below are the specific comments: 1. Temperature performance: The experimental results show that PyTPA performs well at different temperatures.

2. Comparison of advantages and disadvantages: The authors have provided a detailed list of the advantages and limitations of PyTPA, and a comparative table clearly demonstrates its superiority over other dyes.

3. Reaction with non-amine molecules: The authors conducted further selectivity experiments, and the results show that PyTPA does not react with non-amine substances, validating its excellent selectivity.

4. Literature update: The authors have updated the references, incorporating recent relevant studies, which strengthens the timeliness of the literature review.

Overall, the revised manuscript satisfactorily addresses the reviewer comments, with significant improvements. I recommend the manuscript be accepted for publication.

Response: Thank you so much for your constructive comments. It is our pleasure to carry out several experiments that have improved our manuscript significantly. Thank you so much.

Reviewer #4 (Remarks to the Author):

The authors had addressed all comments. While there is still work to do, the papers seems OK to be published at this time. I would state the work is of scientific value but is more incremental than innovative considering there are countless studies on organic amine sensors.

Response: Thank you so much for your positive comments. We will keep your points in mind for our continuing inventions. Thank you so much.